# Transcatheter Aortic Valve Replacement vs. Surgical Aortic Valve Replacement for Long-Term Mortality Due to Stroke and Myocardial Infarction: A Meta-Analysis during the COVID-19 Pandemic

**DOI:** 10.3390/medicina59010012

**Published:** 2022-12-21

**Authors:** Alexandru Cristian Ion, Liviu Ionut Serbanoiu, Elena Plesu, Stefan Sebastian Busnatu, Catalina Liliana Andrei

**Affiliations:** 1Faculty of Medicine, Department 5, Cardiology, Carol Davila University of Medicine and Pharmacy, 020021 Bucharest, Romania; 2Cardiology Department, Bagdasar Arseni Emergency Hospital, 041915 Bucharest, Romania

**Keywords:** transcatheter aortic valve replacement (TAVR), surgical aortic valve replacement (SAVR), stroke, stroke mortality, COVID-19 pandemic

## Abstract

*Background and objectives:* One of the leading causes of mortality and morbidity in people over the age of 50 is stroke. The acceptance of transcatheter aortic valve replacement (TAVR) as a treatment option for severe symptomatic aortic stenosis (AS) has increased as a result of numerous randomized clinical trials comparing surgical aortic valve replacement (SAVR) and TAVR in high- and intermediate-risk patients, showing comparable clinical outcomes and valve hemodynamics. *Materials and Methods***:** An electronic search of Medline, Google Scholar and Cochrane Central was carried out from their inception to 28 September 2022 without any language restrictions. *Results***:** Our meta-analysis demonstrated that, as compared with SAVR, TAVR was not linked with a lower stroke ratio or stroke mortality. It is clear from this that the SAVR intervention techniques applied in the six studies were successful in reducing cardiogenic consequences over time. *Conclusions***:** A significantly decreased rate of mortality from cardiogenic causes was associated with SAVR. Additionally, when TAVR and SAVR were compared for stroke mortality, the results were nonsignificant with a *p* value of 0.57, indicating that none of these procedures could decrease stroke-related mortality.

## 1. Introduction

According to the 2019 Global Burden of Disease (GBD) data [1], stroke was one of the leading causes of disability-adjusted life years (DALYs) among people 50 years of age or older, from 1990 to 2019; however, in the United States, heart disease claimed the lives of nearly 697,000 people in 2020 alone, accounting for one in five fatalities [2].

Aortic valve stenosis may lead to atrial and ventricular remodeling and predispositions to atrial fibrillation (AF) and may also be an independent risk factor of ischemic stroke. The information on stroke incidence in patients with aortic stenosis (AS) are very few. In a Danish retrospective cohort study, which compared the stroke ratios in patients with AS vs. controls, the incidence of stroke was 13.3/1000 person year (PY) among the controls compared with 30.4/1000 PY in patients with AS, corresponding to a hazard ratio of 1.31 (95% CI, 1.28–1.34) [3]. Even if the relative risk and incidence rate were higher in the AS arm for all the age groups, the relative risk was higher in younger individuals (age group, 18–45 years: hazard ratio, 5.94 [95% CI, 4.10–8.36]) [3]. In patients with AS above 65 years of age, the risk of ischemic stroke was lower after aortic valve replacement (AVR) (30.3 versus 19.6/1000 PY before and after AVR), but no analysis on surgical aortic valve replacement (SAVR) vs. transcatheter aortic valve replacement (TAVR) was conducted [3]. Regarding AF cohort patients, the incidence of ischemic stroke was 1.5 times higher when AS was present (33.0/1000 PY versus 49.9/1000 PY) [3]. Meanwhile, regarding AF, the presence of this medical condition in patients with TAVR for AS is associated with a different prognosis, according to the subtype of AF (paroxysmal, nonparoxysmal, new onset) [4]. While paroxysmal AF was not associated with higher 30-day mortality rate post-TAVR, the nonparoxysmal AF was associated with a higher overall mortality rate (hazard ratio: 1.61, 95% confidence interval: 1.35–1.92; *p* < 0.001), but not with a higher 30-day mortality rate [4]. The new-onset AF demonstrated higher 30-day mortality rates (hazard ratio: 2.76, 95% confidence interval: 1.25–6.09; *p* = 0.010) and higher overall mortality [4].

Another study compared the stroke predictors in patients with AS, the prognostic implications of stroke and how AVR influences the predicted outcomes [5]. Rates of stroke were 5.6 versus 21.8 per 1000 PY pre- and post-AVR. Atrial fibrillation (hazard ratio [HR], 2.7; 95% confidence interval [CI], 1.1–6.6), CHA2DS2-VASc score (HR 1.4 per unit; 95% CI, 1.1–1.8), diastolic blood pressure (HR, 1.4 per 10 mm Hg; 95% CI, 1.1–1.8), and the AVR with concomitant coronary artery bypass grafting (HR, 3.2; 95% CI, 1.4–7.2, all *p* ≤ 0.026) was independently associated with stroke [5]. An incidence of stroke predicted death (HR, 8.1; 95% CI, 4.7–14.0; *p* < 0.001) [5]. These finding further support the importance of AF as a main mechanism of stroke in patients with AS.

Referring to myocardial infarction (MI) in patients with AS, the prevalence of coronary artery disease (CAD) in patients undergoing SAVR has been shown to increase with both age and the presence of valve calcification [6,7]. This was demonstrated in a large Swedish registry where coronary artery bypass grafting (CABG) occurred simultaneously with SAVR in 7.2% of patients aged ≤ 50 years, 30.2% of patients aged between 51 and 60 years, 41.2% of patients aged 61–70 years and 51.2% of patients aged ≥ 71 years (f). In a study of 388 patients (mean age 72 years) with aortic valve calcification who underwent coronary angiography, there was a significant association between aortic valve calcification and significant CAD. Thus, aortic valve calcification can serve as a marker for the atherosclerosis of the coronary arteries [7]. Narrowing of the epicardial coronaries or the dysfunction of coronary circulation may lead to an abnormal coronary flow reserve (CFR), even in the absence of angiographically proven atherosclerotic disease. Patients with AS and angiographically normal coronary arteries were shown to have decreased CFR, which decreases the capacity of coronary circulation to increase flow to match myocardial oxygen demand. This impairment of CFR is certainly one of the key elements responsible for myocardial ischemia in patients with AS and may contribute to the development of symptoms, left ventricle (LV) dysfunction and adverse outcomes [8]. The mechanisms underlying the reduction of CFR in patients with AS remain unclear, as concentric LV hypertrophy was previously believed to be the major cause of the reduction in CFR in patients with AS, but recent data suggest that the abnormally high myocardial oxygen consumption (due to LV hypertrophy) induced by AS may be the key factor. In fact, reduced CFR correlates better with hemodynamic indexes of AS severity (valve effective orifice area and transvalvular pressure gradient) than with LV mass [9].

On stroke rates post-TAVR, in a large study assessing the late (>30-day) stroke incidence post-TAVR, patients were followed for a median of 24.5 (IQR 11.4–42.7) months, yielding a total of 10,467 PY for the study [10]. In total, 235 stroke patients were identified for the analysis. The mean annual incidence of stroke during the follow-up varied between 2.00% (95% CI 1.54–2.46%) and 3.12% (95% CI 1.75–4.48%) compared with the standardized incidence, which ranged between 1.46 and 1.93% [10]. In a univariable analysis, eGFR  <  30 mL/min/1.73 m^2^, age, male sex, history of stroke, diabetes, mildly reduced LVEF, peripheral vascular disease and dialysis after procedure were predictors of late stroke, whereas a valve-in-valve (TAVR in SAVR) procedure was associated with a lower risk for stroke. In a multivariable analysis, eGFR  <  30 mL/min/1.73 m^2^ (HR 2.07 [96% CI 1.39–3.11]), diabetes (1.59 [1.19–2.12]), history of stroke (1.48 [1.07–2.05]), age (HR per year 1.03 [1.00–1.05]) and male sex (1.28 [0.99–1.67]) were predictors of late stroke, whereas a valve-in-valve procedure (0.09 [0.01–0.62]) was associated with a lower incidence of stroke [10].

In general, early stroke (within the first 7 days) post-TAVR is broadly considered to be related to the procedure, because of particle embolization. These particles are composed of tissue fragments from the aortic valve, aorta, left ventricular myocardium and thrombus, as shown in studies using embolic protection devices during TAVR [11]. A potential explanation for a delayed (up to 7 days) diagnosis of a stroke causally related to the procedure might be a lack of early imaging and the prolonged time for thrombus formation on the embolized material and subsequent clinical presentation [12].

During the COVID-19 pandemic, there was a significant increase in both age-adjusted and risk-adjusted mortality rates from heart disease and stroke [13]. A cohort study by Sidney et al. [14] showed a 4.1% increase in heart disease deaths and 5.2% increase in stroke deaths when compared with the period 2011–2019. Meanwhile, there was also a 17.6% increase in total age-associated heart disease mortality and a 18.1% increase in deaths from stroke from 2019 to 2020 [14]. These data show the importance of assessing the mortality from stroke and myocardial infarction, especially during the COVID-19 pandemic.

The acceptance of TAVR as a therapeutic option for severe symptomatic AS has been advanced by the results of numerous randomized clinical trials comparing surgery with TAVR in high- and intermediate-risk older adult patients [15,16,17]. TAVR is increasingly used in place of SAVR today [18]. Since the initial TAVR treatment in 2002, TAVR has gained prominence in the field of structural heart disease and is a feasible therapeutic option for AS [19]. It has swiftly become the therapy of choice for a large number of patients with AS [19]. However, there is little information available on the bioprosthetic’s long-term effectiveness and death from stroke and on other cardiogenic causes [20].

When taking into account the TAVR and SAVR incidence during the COVID-19 pandemic, we found that in a cohort study by Glen P. Martin et al. (2021), a rapid and significant drop in TAVR and SAVR activity during the COVID-19 pandemic, especially for elective cases, was noticed. Cumulatively, over the period from March to November 2020, an estimated number of 4989 (95% CI, 4020–5959) cases of aortic stenosis did not receive treatment [21]. While the activity and outcomes for aortic valve replacements (AVRs) have been studied in historical cohorts [22], there is a lack of data in contemporary practice, especially surrounding the impact of COVID-19 from a national perspective.

This issue is also consistent with an increasing waiting time for AVRs [23] and with adverse impacts on outcomes [24]. In a population-based study including TAVR during 2010–2016 by Elbaz-Greener et al. [24], a significant, nonlinear relationship between TAVR wait time and post-TAVR 30-day mortality, as well as 30-day readmission, was noticed.

These unmet problems during the COVID-19 pandemic raised the issue of possible solutions. In a recent mathematical model trying to find a solution for the excess waiting list of patients with AS needing either SAVR or TAVR, the most viable solution was found to be a combination of converting 40% of cases from SAVR to TAVR and increasing capacity by 20%, which would clear the backlog within a year (343 (281–410) days), with 784 (292–1324) deaths while awaiting treatment [25]. This solution was found to be more effective than a higher conversion rate alone or increasing capacity alone [25].

In this meta-analysis, we aimed to investigate whether TAVR was more effective in reducing the mortality due to stroke or any other cardiogenic cause, such as myocardial infarction in the long-term, or whether the conventional method of SAVR was still superior in reducing the mortality rate in high-risk and medium-risk patients. We investigated this by utilizing the data from randomized clinical trials, which provided the data for up to 10 years, especially during the COVID-19 pandemic. Most of the long-term mortality rates were provided for up to 5 years of stroke and cardiogenic outcomes.

## 2. Methods

An electronic search of Medline, Google Scholar and Cochrane Central was carried out from their inception to 28 September 2022 without any language restrictions, using the following search strategy:

(“TAVR”[All Fields] AND (“surgical instruments”[MeSH Terms] OR (“surgical”[All Fields] AND “instruments”[All Fields]) OR “surgical instruments”[All Fields] OR (“surgical”[All Fields] AND “valve”[All Fields]) OR “surgical valve”[All Fields]) AND (“replace”[All Fields] OR “replaceable”[All Fields] OR “replaced”[All Fields] OR “replaces”[All Fields] OR “replacing”[All Fields] OR “replacement”[All Fields] OR “replantation”[MeSH Terms] OR “replantation”[All Fields] OR “replacement”[All Fields] OR “replacements”[All Fields])) AND (“adult”[MeSH Terms] OR “adult”[All Fields] OR “adults”[All Fields] OR “adult s”[All Fields] OR (“aged”[MeSH Terms] OR “aged”[All Fields] OR “elderly”[All Fields] OR “elderlies”[All Fields] OR “elderly s”[All Fields] OR “elderlys”[All Fields])) AND (“mortality”[MeSH Terms] OR “mortality”[All Fields] OR “mortalities”[All Fields] OR “mortality”[MeSH Subheading]) AND (clinicaltrial[Filter] OR randomized controlled trial[Filter])

### 2.1. Translations

Surgical valve: “surgical instruments”[MeSH Terms] OR (“surgical”[All Fields] AND “instruments”[All Fields]) OR “surgical instruments”[All Fields] OR (“surgical”[All Fields] AND “valve”[All Fields]) OR “surgical valve”[All Fields].

replacement: “replace”[All Fields] OR “replaceable”[All Fields] OR “replaced”[All Fields] OR “replaces”[All Fields] OR “replacing”[All Fields] OR “replacment”[All Fields] OR “replantation”[MeSH Terms] OR “replantation”[All Fields] OR “replacement”[All Fields] OR “replacements”[All Fields].

Adults: “adult”[MeSH Terms] OR “adult”[All Fields] OR “adults”[All Fields] OR “adult’s”[All Fields].

elderly: “aged”[MeSH Terms] OR “aged”[All Fields] OR “elderly”[All Fields] OR “elderlies”[All Fields] OR “elderly’s”[All Fields] OR “elderlys”[All Fields].

Mortality: “mortality”[MeSH Terms] OR “mortality”[All Fields] OR “mortalities”[All Fields] OR “mortality”[Subheading] 7822:32:26.

In addition to that, we manually screened the reference list of retrieved trials, review articles and previous meta-analyses to identify any relevant studies. However, only the randomized trial studies and cohort studies were included in our meta-analysis.

### 2.2. Study Selection

#### 2.2.1. Inclusion Criteria

The following eligibility criteria were used to select studies: (a) published randomized controlled trials or cohort studies; (b) the experimental and control populations included in the studies had at least one neurological or cardiogenic outcome reported.

#### 2.2.2. Exclusion Criteria

(a)Any study that was not a trial.(b)Studies over five years old.(c)Studies that did not contain any control or experimental data.(d)Studies that did not report a neurological or cardiogenic cause of mortality.

#### 2.2.3. Data Extraction and Quality Assessment

The articles discovered via the systematic search were imported into EndNote Reference Library, where duplicates were recognized and removed. Only studies that met the previously defined criteria were selected from the remaining papers, which were extensively evaluated by two investigators. All trials were first shortlisted based on the title and abstract, and the whole article was then reviewed to ensure relevancy. Furthermore, any inconsistencies were excluded. The completed trials yielded the following result: stroke and cardiovascular causes such as MI. It was retrieved using an Excel spreadsheet, and all data and values were preserved in the spreadsheet for subsequent study. In addition, the Cochrane collaboration’s risk of bias tool for randomized controlled trials was used to assess the quality of the studies and provide a plot and risk of bias summary for each (Figure 1).

#### 2.2.4. Statistical Analysis

RevMan (version 5.3; Copenhagen: The Nordic Cochrane Centre, The Cochrane Collaboration, 2014) was used for all statistical analyses. The results from the studies were presented as means and standard deviations with 95% confidence intervals (CIs) and were pooled using a random effects model. Forest plots were created to visually assess the results of pooling. Furthermore, a funnel plot was also constructed to evaluate the publication bias in studies, in addition to the risk of bias graph and risk of bias summary chart.

## 3. Results

### 3.1. Results of Literature Search

Using the trials, RCT and 5-year time filters, 594 possible studies on Medline, 22 on Cochrane Central and 4320 on Google Scholar were found after an initial search of the three electronic databases. After the exclusions, 10 trials remained for analysis. Only eight of those that met the criteria were picked after the trials had been thoroughly evaluated in full text (Figure 2).

### 3.2. Study Characteristics

Only eight studies were included in the final meta-analysis. The number of participants amounted to 14,601. Among these 14,601 patients were 6290 participants who were randomly assigned to the TAVR procedure in the stroke outcome, while 8311 patients were randomly assigned to the control arm, which was SAVR in the stroke outcome. In contrast, for the cardiogenic cause, the total number of the patients was 5973. Throughout the six investigations for this outcome, among those 5973 patients, 3024 patients were randomly assigned to the experimental arm, which was TAVR, and 2949 patients to the control arm, which was SAVR.

### 3.3. Results of Meta-Analysis

Eight trials evaluating the effectiveness of TAVR vs. SAVR were included. According to the analysis, in the long term, TAVR did not significantly reduce the incidence of stroke in the experimental population compared with the control population treated by SAVR intervention (Figure 3).

The findings show that all the research studies carried equal weights in the pooling of studies (12.5%) in the stroke outcome, along with a 95% CI of −14.24 [−62.81, 34.33], whereas in the forest plot of the cardiogenic cause, Ito 2020′s research study had the lowest weight (10.3%) and the largest spread among the pooled studies, with a 95% CI of 15.00 [138, 28.62]. The studies’ heterogeneity turned out to be 99% for a cardiogenic cause such as MI, and the findings were significant, with a *p* value of 0.02. This demonstrates that the SAVR intervention procedures used in the six studies were effective in lowering the cardiogenic causes in the long term. SAVR was associated with a significantly lower rate of mortality due to cardiogenic causes. Moreover, when TAVR and SAVR were analyzed for the mortality due to stroke, the results turned out to be nonsignificant, with a *p* value of *p* 0.57, which indicated that the TAVR and SAVR could not assist in preventing stroke in the long term (Figure 4).

Furthermore, when the publication bias of the pooled studies was examined, there was publication bias in the funnel plot for the TAVR and SAVR in stroke as well as in the MI outcome, which was created using RevMan (version 5.3; Copenhagen: The Nordic Cochrane Centre, The Cochrane Collaboration, London, UK 2014) (Figure 5 and Figure 6).

## 4. Discussion

The effective use of TAVR as a significant new treatment for individuals with symptomatic severe AS has been supported by large clinical data. The main goals of the clinical studies were to prove early safety and effectiveness over a period of 1 to 2 years. Numerous 5-year follow-up studies on patients who had TAVR and had severe, high or moderate surgical risk have been published [31,32].

In our meta-analysis, after conducting a rigorous analysis, we found that TAVR is suitable, over SAVR, in patients with intermediate risk. However, in patients with severe aortic stenosis who are at risk of developing stroke mortality, TAVR did not prove to be significantly associated with a lower risk of cardiovascular death at 5 years and beyond, in this meta-analysis of RCTs. Even if TAVR is regarded as a superior and more common procedure than SAVR, it was unable to effectively reduce stroke mortality in patients over a 5-year time analysis. To the best of our knowledge, we believe that this is the meta-analysis that had added trials of greater follow-up terms to assess the performance of TAVR vs. SAVR at up to 10 years of duration. Other studies have proven that when compared with traditional surgery, TAVR has shown to protect the patient against stroke mortality, particularly in the first 30 days following the procedure [26] and maybe even till 1 year [33].

Yake Lou et al. stated in their meta-analysis that for patients with a low-to-intermediate surgical risk, TAVR had at least a comparable clinical impact to SAVR for 2 years after therapy. They also noted that TAVR was linked to a decreased incidence of severe bleeding and stroke [34].

Furthermore, Jørgensen et al. posted study results that showed that there was no statistical difference in the risk for all-cause mortality, stroke or myocardial infarction or in the risk of bioprosthetic valve failure after 8 years of follow-up in patients with severe aortic valve stenosis at low surgical risk who were randomized to TAVI or SAVR [35]. Another study analyzed AVR vs. conservative treatment between June 2015 and September 2020. In this study, 157 patients were randomly assigned to early surgery (*n* = 78) or conservative care (*n* = 79). The overall median follow-up was 32 months, with 28 months in the early surgery group and 35 months in the group receiving conservative care: 39 deaths occurred in all patients—13 in the early surgery group and 26 in the group receiving conservative care. Moreover, 72 patients (92.3%) in the early surgery group underwent SAVR, with a procedure mortality rate of 1.4%. Patients assigned to early surgery had a significantly lower incidence of the primary composite end point than did those in the conservative arm, according to an intention-to-treat analysis (hazard ratio, 0.46 [95% CI, 0.23–0.90]; *p* = 0.02). In secondary endpoints, there was no statistically significant difference, even when trends were in line with the primary outcome, such as all-cause death, initial heart failure hospitalizations, significant bleeding or thromboembolic events [36].

In a retrospective cohort study by MD Andreasen, it was revealed that patients with atrial fibrillation have a significantly increased risk of ischemic stroke when they have aortic valve stenosis [3].

In addition to the above studies, Michael J Mack et al. concluded in their study that the rate of the composite of death, stroke or rehospitalization at 1 year was considerably lower with TAVR compared with surgery among patients with severe aortic stenosis who were at low surgical risk [26].

Jaakkola concluded from an observational study that after transcatheter aortic valve replacement for aortic stenosis, mortality was increased in nonparoxysmal atrial fibrillation and new-onset atrial fibrillation, but not in paroxysmal atrial fibrillation. These results implied that in patients with aortic stenosis undergoing transcatheter aortic valve replacement, nonparoxysmal atrial fibrillation rather than paroxysmal atrial fibrillation may be linked to structural cardiac damage that is significant for prognosis [4].

In Amrane et al., there was no difference in the stroke mortality rate between TAVR and SAVR, during the recovery period (30–120 days) or during the late period (120–365 days) [27]. In our meta-analysis, there was no statistically significant benefit of TAVR compared with SAVR for stroke, over an extended period of time. In the Chen et al. study (PARTNER 2A), which included patients at moderate surgical risk, there was no significant difference between the cardiovascular death rates in the SAVR and TAVR arms (11.8% vs. 10.5%, *p* = 0.42) and no difference in the stroke rate (8.8% vs. 9.2%, *p* = 0.92) or stroke mortality rate between the two groups [28]. This finding is consistent with the findings on high-risk patients. Moreover, Reardon et al. found no significant difference in the cardiovascular mortality at 1 year between TAVR and SAVR (1.7% vs. 2.6%), with similar stroke rates (4.1% vs. 4.3%) [29]. This result also shows that the findings apply to all risk patients, from low to high risk.

Amartya Kundu’s meta-analysis, which included 2953 patients from five studies, noted that TAVR was linked to a similar level of mid-term mortality as SAVR was. A lower likelihood of having acute renal injury, short-term severe hemorrhage and newly onset atrial fibrillation was linked to TAVR. However, TAVR was linked to an increased chance of needing to implant a permanent pacemaker. The risk of myocardial infarction, stroke, endocarditis or aortic valve reintervention was not significantly different between the two groups [37].

In another study by Leon et al., the low-risk patients with severe AS undergoing TAVR showed a lower stroke rate and cardiovascular mortality rate at 2 years compared with SAVR, but these results were not statistically significant at 2 years (death: TAVR 2.4% vs. surgery 3.2%; *p* = 0.47; stroke: TAVR 2.4% vs. surgery 3.6%; *p* = 0.28). Meanwhile, the valve thrombosis risk was higher in TAVR patients compared with SAVR [30].

During the COVID-19 pandemic, the decrease in TAVR and SAVR incidence was associated with higher mortality in patients with severe AS waiting for intervention. In addition, the future management of aortic valve diseases during the COVID-19 pandemic is likely to see an increase in virtual assessments and consultations and the expanded use of TAVI as a treatment option [38]. The paper by a group from the Royal Papworth Hospital at the University of Cambridge, UK, should serve as concrete proof that SAVR remains an excellent option with favorable outcomes in the treatment of AS [39]. These findings are related to other research conducted during the COVID-19 pandemic on the bleeding risk of antithrombotic therapy [40].

However, SAVR showed more promising results in medium- to high-risk individuals regarding mortality due to myocardial infarction in the long term. In order to assess the effectiveness of TAVR and SAVR in reducing the risk of stroke 5 to 10 years after procedures, this meta-analysis found excellent randomized controlled studies. Data from trials that lasted up to 10 years and 5 years on average were assessed to determine the effectiveness of the TAVR and SAVR techniques. In contrast to a study conducted by Kolte in 2019, where the author and colleague conducted a research study assessing the mortality at 1 year duration when the patients had undergone either TAVR or SAVR, the results showed significant and promising results in favor of TAVR in both stroke and a cardiogenic cause of death [33]. We found that TAVR was not associated with a reduction in stroke-related mortality compared with SAVR in the long term. Furthermore, our findings were concordant with a study conducted in 2018 that highlighted that TAVR generates fewer stroke incidences compared with SAVR (3.3% vs. 5.4%; *p* = 0.031) [32] in the short term, but as we crossed the threshold of 5 years and added the data from a study conducted in Finland that followed the results for 10 years, we found that TAVR did not shine in causing fewer strokes than SAVR did [19].

## 5. Limitations

In terms of constraints, only eight research papers were eligible for inclusion in the analysis for the stroke outcome and only six for the MI outcome. A higher number of research studies might give additional information about the effectiveness of TAVR vs. SAVR in reducing long-term mortality due to neurological causes such as stroke and cardiac causes such as MI. Furthermore, after constructing the funnel plot, we assessed it and observed a strong small study publication bias in our meta-analysis, which may be a limitation of our work. In addition to the previously mentioned problems, the studies had a high degree of heterogeneity. As more studies on this subject emerge, a separate meta-analysis regarding patients with different preintervention risks may be assessed, as their prognostic may be different in patients with low vs. medium or high risk.

## 6. Future Implications

Our findings might open a path for further research into the utility of TAVR and SAVR. Furthermore, our results may be the starting point for further analysis on the potential downsides of these procedures and for making changes to assign the surgery considered most suitable for the patient.

## 7. Conclusions

On the basis of the research studies included in our meta-analysis, it is obvious that TAVR is not a viable treatment option to prevent late stroke mortality in patients over 5 years post-surgery. Furthermore, SAVR is better at reducing the cardiogenic cause of mortality in patients than TAVR is over the same duration.

## Figures and Tables

**Figure 1 medicina-59-00012-f001:**
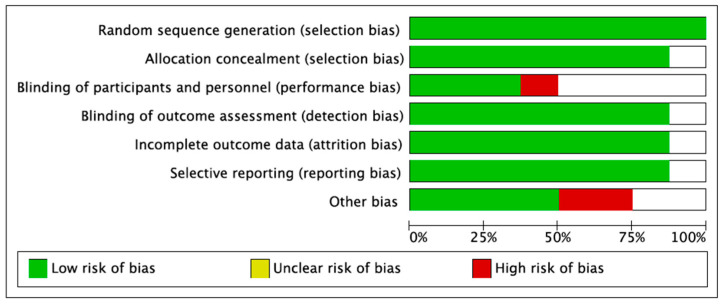
Risk of bias assessment.

**Figure 2 medicina-59-00012-f002:**
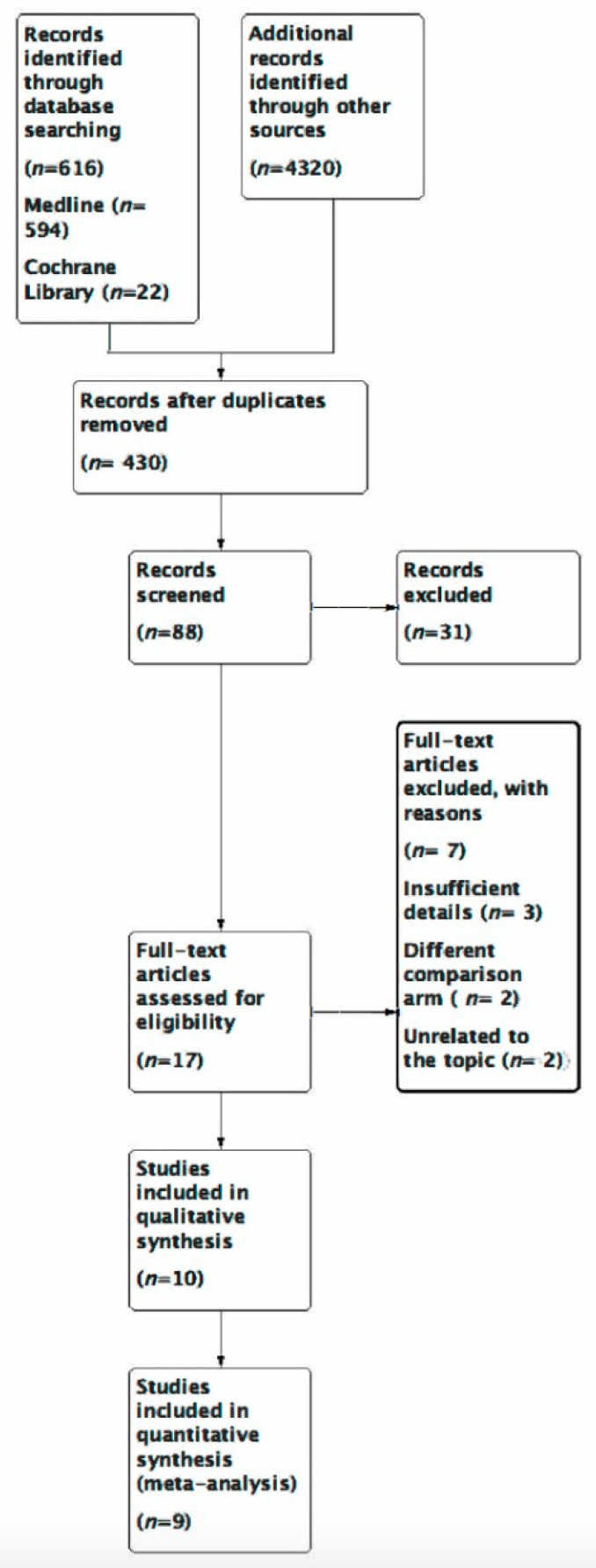
Prisma flow diagram.

**Figure 3 medicina-59-00012-f003:**
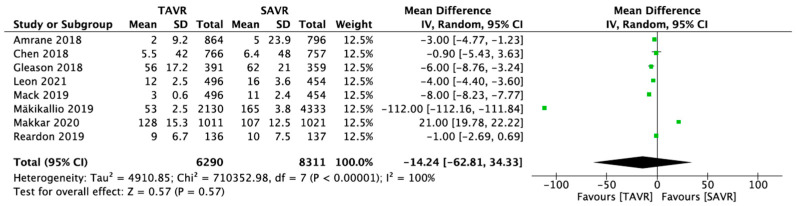
Forest plot of the pooled studies for stroke [18,19,20,26,27,28,29,30].

**Figure 4 medicina-59-00012-f004:**
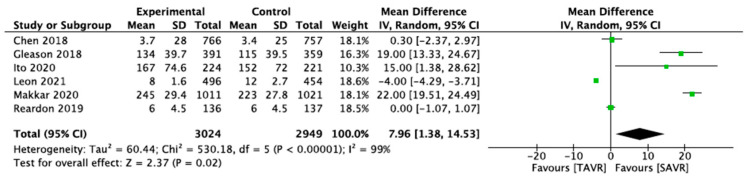
Forest plot of the pooled studies for cardiogenic cause (MI) [18,20,28,29,30].

**Figure 5 medicina-59-00012-f005:**
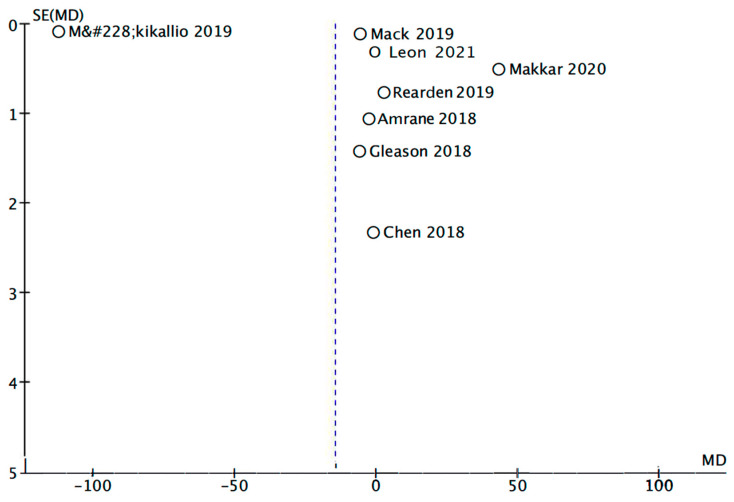
Funnel plot for publication bias (stroke) [18,19,20,26,27,28,29,30].

**Figure 6 medicina-59-00012-f006:**
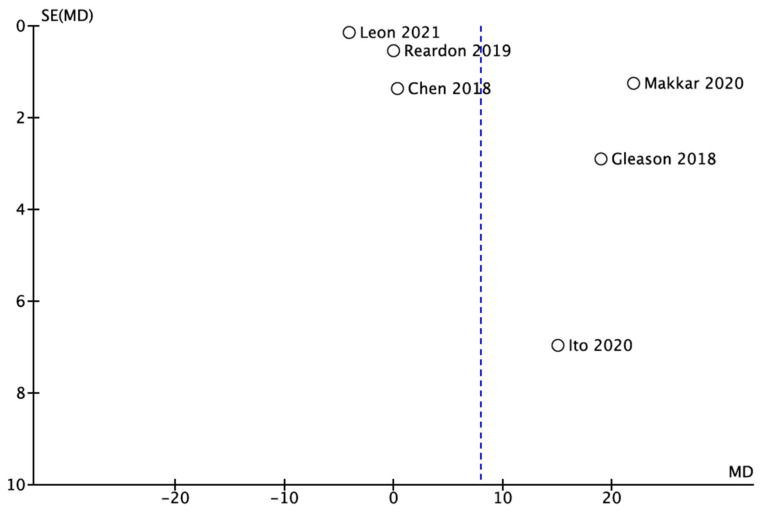
Funnel plot for publication bias (cardiogenic cause) [18,20,28,29,30].

## Data Availability

Not applicable.

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
