# Peer review of "Transcatheter Aortic Valve Replacement vs. Surgical Aortic Valve Replacement for Long-Term Mortality Due to Stroke and Myocardial Infarction: A Meta-Analysis during the COVID-19 Pandemic"

_medicina, 2022, doi:10.3390/medicina59010012_

Round 1

Reviewer 1 Report

In this meta-analysis, the authors compared the long term mortality after TAVR vs SAVR due to stroke and MI. They concluded that TAVR is a viable treatment option to prevent early stroke mortality in patients of low and intermediate risk compared to SAVR. However, it is not viable in the long-term at 5 or over years. SAVR is better at reducing the cardiogenic cause of mortality .

This is very important topic. All the sections of the paper are well written except the result section. I think that this section should be re- written. It is difficult to understand the details. For example: It is written that "6290 stroke patients were randomly assigned to the experimental arm of the research, while 8311 patients were randomly assigned to the control arm. While for the cardiogenic reason, 3024 patients were randomly assigned to the experimental arm and 2949 patients  to the control arm throughout the six investigations"  . Authors should clarify this data. They should define research arm? control arm?

Author Response

First of all, thank you for the opportunity to make our manuscript better. Indeed this sectionseemed to be a little confusing. We have redone it to clarify it in clear detail to make it easier forthe reader to understand the division of participants into the experimental and control arms. If there's any other correction that needs to be made, we would appreciate that too

Reviewer 2 Report

The authors aimed at investigating whether SAVR or TAVR for severe aortic stenosis are protective against stoke or myocardial infarction. The subject is of great importance for the clinicians.

However, some major issues are to discuss:

1.       The title is somehow twisted, as all the studies including procedures for aortic stenosis repair have as primary endpoint all-cause mortality and as secondary outcomes bleeding, stroke, myocardial infarction (MI), and other complications of aortic-valve replacement. As for COVID19, it had a profound inflammatory effect on the blood vessels, both in the arterial and venous territory with a high incidence of stroke and MI. The major problem during COVID19 was the long period (months) of blockage of the medical system during which performing procedures such as TAVR or SAVR was difficult. The meta-analysis does not and cannot evaluate long term effects of COVID19 on patients with aortic stenosis and TAVR or SAVR, because the data in the randomised control trials published in 2018 and 2019 long forego COVID19. In this respect, I would recommend that COVID19 should be removed from the title and from the text.

2.       The introduction should be more structured to underline the importance of the subject. There are 2 lines to follow:

a.       Stroke: its prevalence in patients with severe aortic stenosis and the possible mechanisms, including atrial fibrillation

b.       Myocardial infraction: its prevalence in patients with severe aortic stenosis and the possible mechanisms, different from those responsible for stroke

c.       Repairing procedures of aortic valve could cause stroke ore MI (complication rates and possible mechanisms should be mentioned).

The authors should clearly state in the objective if they focus only on the stroke and MI as a complication of AS. If so, how they could be prevented by the interventional procedure. Please specify how could one differentiate between stroke caused by severe aortic stenosis and stroke as o complication of TAVR/SAVR. By which means did the selected clinical studies made those differences? A large study of the DANISH group might be of help in this case (https://www.ahajournals.org/doi/epub/10.1161/STROKEAHA.119.028389 ).

3.       The result section does not comprise some valuable articles such as Jorgensen et al. at https://doi.org/10.1093/eurheartj/ehab375 who showed no significant differences in the risk for all-cause mortality, stroke, or myocardial infarction, as well as the risk of bioprosthetic valve failure after 8 years of follow-up. Could the authors explain why this trial failed to qualify in the meta-analysis?

Does the presence of AF was taken into account when the risk of stroke was evaluated?

4.       The discussion section should start with highlighting 3-4 of the main findings of these meta-analysis and continues with comparing the results with other similar studies. The authors claimed that this might be the only published meta-analysis regarding stroke and MI after TAVR and SAVR. As a matter of fact, there are some other meta-analyses published in the past 2 years:

a.       Lou et al. https://doi.org/10.3389/fcvm.2020.590975

b.       Malik et al https://doi.org/10.11909/j.issn.1671-5411.2020.01.005  

c.       Zhang et al. https://doi.org/10.1097/SLA.0000000000003906  

d.       Kundu et al. https://doi.org/10.1016/j.carrev.2019.08.009

e.       Al-Abdouh A. https://doi.org/10.1016/j.carrev.2019.08.008

Interestingly, the published data from those meta-analysis are contrasting. I would kindly invite the authors to comment upon these meta-analyses and the differences found.

5.     The conclusions should be revised to clearly respond to the proposed objective.

6. The overall text should be checked for spelling errors. Some observations were made directly on the text. Please provide the requested changes.

Yours sincerely,

Author Response

  1. Thank for this suggestion. We changed the title by emphasizing the fact that our analysis was made during COVID 19 pandemic. Meanwhile, some of the included studies were publicated during COVID 19 pandemic, even if these studies didn’t include COVID 19 patients. The direct impact of COVID 19 on patient’s prognosis was not the objective of our study, as we didn’t analyse the difference between SARSCOV2 inected patients and patients without this pathology. The impact of COVID 19 pandemic on TAVR and SAVR procedures was largely analysed both in introduction and discussion sections.
  2. Indeed this section seemed to be a little confusing. We have redone it to clarify it in clear detail to make it easier for the reader to understand. If there's any other correction that needs to be made, we would appreciate that too.

We included the data regarding the incidence of stroke and myocardial infarction in patients with AS, taking into account the main causes for these complications. As stated by the respected reviewer, the mechanism are completely different regarding stroke an myocardial infarction. We also included data regarding the incidence of myocardial infarction and stroke post TAVR and SAVR.

 Respected reviewer pointed out and inquired why we haven't included the study of the DANISH group. We would like to inform the respected reviewer that the Danish study wouldn't have made it to the eligible studies because of the search strategy and filters used since it didn’t have any quantitative data specifically regarding TAVR and SAVR surgery. However we thank you for presenting this study to us and we have talked about it in the discussion portion. Thank you once again for that.

  1. Thank you for highlighting this concern and presenting your valuable feedback towards our manuscript. It seems like it would have been missed in our search strategy, however we have added a little paragraph about it in the discussion to highlight the study too.

Point 2: The presence of AF is talked about in the discussion however it was not taken into consideration when the risk of stroke was evaluated.

  1. Respected reviewer, thank you for guiding us and pointing our shortcomings and giving a chance to enhance the manuscript. With regard to your concern about the point 4 in which you have provided the 5 studies to compare and contrast, we have made the necessary comments and compare and contrasted the studies that you have provided to us.

The published data is contrasting to the other studies that have been conducted, the sole reason is that we included the eligible study from Finland with 10 year follow up in which the amount of subjects were greater and the results showed divergence from TAVR performance in the long run, we felt it is a crucial study with crucial findings and we couldn't exclude that, thus the results turned out different. Otherwise TAVR is superior in the medium time-frame.

Furthermore the meta-analysis have been taken into consideration in the discussion sections. Once again thank you for that.

  1. First of all, thank you for the opportunity to make our manuscript better. The conclusion has been revised, concised and redone to clearly conclude the objective.

  1. Thank you for pointing the spelling and grammatical mistakes, The text has been extensively checked to get rid of even the slightest grammatical and spelling mistakes. Thank you for your kind guidance.

Round 2

Reviewer 1 Report

The authors addressed my suggestions

Reviewer 2 Report

Dear authors,

It was a pleasure to participate to the review process of this manuscript. 

All observations were taken into account and the manuscript has been improved considerably.

English language and style are fine, but spell check for minor spelling errors is still required. 

Yours sincerely,